# Clinical Presentation of the c.3844T>C (p.Trp1282Arg, W1282R) Variant in Russian Cystic Fibrosis Patients

**DOI:** 10.3390/genes11101137

**Published:** 2020-09-27

**Authors:** Nika V. Petrova, Nataliya Y. Kashirskaya, Stanislav A. Krasovskiy, Elena L. Amelina, Elena I. Kondratyeva, Andrey V. Marakhonov, Tatyana A. Vasilyeva, Anna Y. Voronkova, Victoria D. Sherman, Evgeny K. Ginter, Sergey I. Kutsev, Rena A. Zinchenko

**Affiliations:** 1Research Centre for Medical Genetics, Russian Federation, Moskvorechie St., 1, 115522 Moscow, Russia; npetrova63@mail.ru (N.V.P.); sa_krasovsky@mail.ru (S.A.K.); elenafpk@mail.ru (E.I.K.); marakhonov@gmail.com (A.V.M.); vasilyeva_debrie@mail.ru (T.A.V.); voronkova111@yandex.ru (A.Y.V.); tovika@yandex.ru (V.D.S.); ekginter@mail.ru (E.K.G.); kutsev@mail.ru (S.I.K.); renazinchenko@mail.ru (R.A.Z.); 2Pulmonology Research Institute under FMBA of Russia, Orekhoviy boulevard, 28, 115682 Moscow, Russia; eamelina@mail.ru

**Keywords:** cystic fibrosis, c.3844T>C (p.Trp1282Arg; W1282R) variant, Russian CF patients, clinical presentation

## Abstract

The goal was to study the phenotypic manifestations of c.3844T>C (p.Trp1282Arg, W1282R) variant, a CF-causing mutation, in patients from the Russian Federation. Clinical manifestations and complications (the age at CF diagnosis, sweat test, pancreatic status, lung function, microbial infection, body mass index (BMI), the presence of meconium ileus (MI), diabetes, and severe liver disease) were compared in four groups: group 1—patients carrying c.3844T>C and severe class I or II variant in trans; group 2—3849+10kbC>T/F508del patients; group 3—F508del/F508del patients; and group 4—patients with W1282R and “mild” variant in trans. Based on the analyses, W1282R with class I or II variant in trans appears to cause at least as severe CF symptoms as F508del homozygotes as reflected in the early age of diagnosis, high sweat chloride concentration, insufficient pancreatic function, and low lung function, in contrast to 3849+10kbC-T/F508del compound heterozygotes having milder clinical phenotypes. The W1282R pathogenic variant is seemed to lead to severe disease phenotype with pancreatic insufficiency similarly to the F508del homozygous genotype.

## 1. Introduction

Cystic fibrosis (CF; OMIM #219700) is an autosomal recessive hereditary disease in which the molecular genetic basis is a mutation of the *CFTR* gene (OMIM *602421). Cystic fibrosis is characterized by a broad clinical presentation [1,2,3]. Differences in CF severity can be explained primarily by the variety of pathogenic variants and genotypes of CF patients. In-depth studies on the consequences of different mutations on CFTR protein have simplified functional pathogenic mechanisms to six kinds of defects which led to the division of pathogenic *CFTR* variants into six classes: class 1—those affecting protein synthesis due to stop-codon (nonsense) mutations; class 2—those affecting CFTR protein processing due to protein misfolding; class 3—those affecting the activation of ion transport function; class 4—those reducing the number of chloride ions transported through the pore channel; class 5—those reducing the amount of wild-type CFTR protein at the plasma membrane; class 6—those affecting the stability and/or anchoring of CFTR protein at the plasma membrane. To date, more than 2000 different variants of the *CFTR* gene sequence have been identified, and about 20% of these are considered to be disease causative (pathogenic) [1,2,3,4,5].

The CF frequency in European countries is 1 per 2500–4500 newborns on average, and in Russia it is about 1 per 10,000 newborns [6,7]. According to the Russian Cystic Fibrosis Patient Registry 2018 (RCFPR), more than 200 pathogenic variants of the *CFTR* gene have been identified in the Russian Federation, the most frequent being eleven variants: c.1521_1523delCTT (p.Phe508del, F508del)—53.05%; c.54-5940_273+10250del21kb (p.Ser18Arg*fsX16, CFTRdele2,3)—6.09%; c.274G>A (p.Glu92Lys, E92K)—3.04%; c.3718-2477C>T (3849+10kbC-T)—2.38%; c.2012delT (p.Leu671X, 2143delT)—2.11%; c.2052_2053insA (p.Gln685ThrfsX4, 2184insA)—1.84%; c.3846G>A (p.Trp1282X, W1282X)—1.75%; c.1545_1546delTA (p.Tyr515X, 1677delTA)—1.77%; c.3909C>G (p.Asn1303Lys, N1303K)—1.55%; c.1624G>T (p.Gly542X, G542X)—1.48%; and c.413_415dupTAC (p.Leu138dup; L138ins)—1.35% [7]. The spectrum and frequency of pathogenic variants in the *CFTR* gene vary significantly in different populations and ethnic groups and differ from the distribution in the all-Russian sample of CF patients. For example, in the North Caucasus Federal district, three variants are most frequent: F508del (25.0%); 1677delTA (21.5%); and W1282X (17.2%), and in the Volga Federal district the most frequent variants are: F508del (50.5%); E92K (8.7%); and CFTRdele2,3 (5.0%) [7]. The study of *CFTR* gene variant spectra in different ethnic groups in the Russian Federation revealed a high proportion of W1282X variant (88%) in Karachay (in the Karachay–Cherkess Republic of the Russian North Caucasus) [8], 1677delTA (81.5%) and E92K (12.5%) in Chechens [9], and E92K (55%) in Chuvash [10]. The second most common variant in Chuvash CF patients is F508del (30%) [10]. In regions with a predominantly Russian population, the most frequent are the F508del (49–65%) and CFTRdele2,3 (2–7%) variants [7,11]. A number of variants that are relatively frequent in Russian patients are not found in the other regions of the world [12,13,14], e.g., L138ins [11,15], W1282R [7,11], and 3272-16T>A [11,16]. Their frequencies according to the RCFPR 2018 database are 1.35%, 0.52%, and 0.34%, respectively [7].

In recent decades, research efforts have focused on developing treatment approaches that take into account the specificity of the molecular consequences of specific *CFTR* variants [17,18]. The study of phenotypic manifestations of all variants of the *CFTR* gene can contribute to the optimization of treatment methods and genetic counseling.

The purpose of this study was to clarify the nature of clinical consequences and features of phenotypic manifestations in CF patients from Russia who carry the c.3844T>C (p.Trp1282Arg, W1282R) variant.

## 2. Materials and Methods

Variant c.3844T>C (p.Trp1282Arg, W1282R) is included in the panel of common mutations that are routinely tested in the Russian population as the first step of molecular analysis of CF [11,14,19]. Molecular genetic testing for the c.3844T>C (p.Trp1282Arg, W1282R) variant is performed by restriction analysis. Modified primers have been designed: W1282R_F 5′-GATTCAATAACTTTGCAACCG-3′, W1282R_R 5′-TATGAGAAAACTGCACTGGA-3′. A 198 bp fragment is amplified. After restriction by endonuclease *Bsh*1236I (Fermentas, Vilnius, Lithuania), two fragments (178 and 20 bp) are formed in the case of the mutant allele 3844C. The normal allele 3844T does not contain a restriction site and the fragment remains uncut (198 bp). Sanger sequencing to confirm the presence of the mutant variant was carried out in the Laboratory of Genetic Epidemiology of the Research Centre for Medical Genetics.

To confirm the heterozygous compound status of a patient’s variants, the parents’ genotypes were determined. The absence of other pathogenic variants in cis with W1282R was confirmed by Sanger sequencing the coding sequence of the *CFTR* gene and analysis of large rearrangements by the MLPA method.

To assess the clinical consequences of the c.3844T>C (p.Trp1282Arg, W1282R) variant, data from the RCFPR for the years 2012–2018 were analyzed. The following variables were taken into account: the patient’s age at the last examination, the age of diagnosis, the sweat test level (sweat chloride, mmol/l), body mass index (BMI; kg/m^2^), spirometric indices of the lungs (forced expiratory volume in 1 s (FEV_1_, % predicted) and forced vital capacity (FVC; % predicted)), chronic colonization of the bronchopulmonary system by microorganisms (*Staphylococcus aureus*, *Pseudomonas aeruginosa*, *Burkholderia cepacia* complex, *Achromobacter xylosoxidans*, *Stenotrophomonas maltophilia*, non-tuberculosis mycobacteria, Gram-negative microflora), pancreatic insufficiency (fecal elastase 1 (<200 μg/g)), complications (meconium ileus, liver cirrhosis (with/without hypertension syndrome), CF-associated diabetes, and allergic bronchopulmonary aspergillosis (ABPA)). Diagnostic criteria of manifestations and complications of CF were used according to the Variables and Definitions of the ECFS Patient Registry (https://www.ecfs.eu/projects/ecfs-patient-registry/Variables-Definitions). 

Phenotype–genotype relationships were studied by comparing three groups of CF patients with different genotypes: group 1—W1282R and F508del, CFTRdele2,3, 1898+1G->A, 2184insA, 3821delT, 541delC, 712-1G->T or W1282X in trans (31 persons); group 2—3849+10kbC-T/F508del (50 persons); and group 3—F508del/F508del (100 persons). Groups 2 and 3 consisted of patients corresponding by sex and most suitable for age for patients of group 1. We also analyzed the features of the clinical manifestation of the disease in nine patients carrying the c.3844T>C (p.Trp1282Arg, W1282R) variant in a compound heterozygous condition with pathogenic variants of the *CFTR* gene associated with the preservation of residual pancreatic function (group 4), and compared them with group 1.

The patients or their parents signed informed consent for participation in the study. The research protocol was approved by the Ethical Committee of the Research Centre for Medical Genetics (115522, Moscow, Moskvorechie St., 1, Russian Federation, Protocol No. 17/2006 of 02.02.2006).

The functional effect of c.3844T>C (p.Trp1282Arg, W1282R) was predicted using the PolyPhen-2 tool available online: http://genetics.bwh.harvard.edu/pph2/ [20]. Further, 3D modeling was performed using the Missense 3D bioinformatic tool available online at http://www.sbg.bio.ic.ac.uk/~missense 3d [21].

Statistical analysis was performed using the STATISTICA v.8.0 program (StatSoft Inc., Tulsa, OK, USA). To compare categorical variables, we used the Fisher test, and the Mann–Whitney *U* test was used for quantitative variables. The significance level *p* ≤ 0.05 was considered significant.

## 3. Results and Discussion

The c.3844T>C (p.Trp1282Arg, W1282R) variant was first described by the Russian researchers T. E. Ivashchenko and co-authors in 1993 in a Russian CF patient [22]. At the same time, this variant was registered in the Cystic Fibrosis Database CFTR1 [4]. The first clinical presentation of eight Russian CF patients carrying the W1282R variant was published in 2013. Krasovsky S. A. and co-authors showed that the W1282R variant could be attributed to “severe” mutations associated with pancreatic insufficiency and severe lung disease [23]. Currently, in addition to being in the CFTR1 database [4], the W1282Rvariant is included in the ClinVar [24] and ExAC [25] databases. ClinVar considers the W1282R variant as a variant with an unknown functional value [24]. The W1282R variant is not listed in the CFTR2 database [5].

The W1282R variant is one of the most common types—missense mutations. As a result of the c.3844T>C substitution, a tryptophan residue is changed to an arginine residue at position 1282 in the protein molecule sequence (p.Trp1282Arg, W1282R).

This variant is located in the second nucleotide-binding domain (NBD2). Both NBDs contain several highly conserved motifs that are thought to be involved in ATP binding and hydrolysis. Site-directed mutagenesis in these motifs has indicated that ATP binds to both NBDs to control the gating of the channel [4]. PolyPhen-2 [20] (the tool for prediction of functional effects of human nsSNPs) predicts this mutation to be probably damaging, with a score of 1.000 (sensitivity: 0.00; specificity: 1.00).

The Missense 3D bioinformatic tool [21] was used to simulate the tertiary structure (3D) of a CFTR molecule containing an arginine residue (R, Arg) at position 1282 instead of the tryptophan residue (W, Trp) (Figure 1). The p.Trp1282Arg (W1282R) substitution replaces a buried hydrophobic uncharged residue (TRP, RSA 0.8%) with a hydrophilic charged residue (ARG, relative solvent accessibility (RSA) 1.2%). Also, as predicted, the p.Trp1282Arg (W1282R) substitution does not alter the secondary structure ‘H’ (4-turn helix) and does not result in complete disruption of all side-chain/side-chain H-bond(s) and/or side-chain/main-chain bond(s) bonds formed by a buried tryptophan residue (RSA 0.8%), and it leads to expansion of the cavity volume by 29.16 Å^3^.

The RCFPR provides information about 42 patients carrying the W1282R variant from 40 unrelated families. The relative frequency of the W1282R variant was 0.55% of all identified *CFTR* gene mutant alleles in the Russian population [7]. The average age of patients carrying the W1282R variant at the time of the last examination was 15.36 ± 1.96 years (0.73—60.06). The sex ratio was 0.45: 0.55 (19 males: 23 females). The average age at diagnosis was 4.95 ± 1.39 years (0.08—39.60). In all 42 patients, the W1282R variant was found in a compound heterozygous state with other *CFTR* gene variants. Sixteen different genotypes were identified and are presented in Table 1. The most frequent genotype, W1282R/F508del, was found in 19 of the CF patients (45.2%).

Phenotypic correlations were studied in a group of patients carrying the W1282R variant (group 1) compared to two groups of patients with different genotypes whose clinical presentation is well known and significantly different. In group 1 there were 31 patients carrying the W1282R and class I or II variant in trans (F508del, CFTRdele2.3, 1898+1G->A, 2184insA, 3821delT, 541delC, 712-1G->T, or W1282X) (Table 1).

In general, patients homozygous for class I–III mutations exhibit a phenotype associated with pancreatic insufficiency, higher frequency of meconium ileus, premature mortality, earlier and more severe deterioration of lung function, higher incidence of malnutrition and severe liver disease. Class IV–V mutations are usually associated with milder lung disease, older age at death, pancreatic sufficiency. Class IV–V mutations are phenotypically dominant when occurring in combination with class I–III mutations [26]. 

For group 2, 50 patients with compound heterozygous genotype 3849+10kbC-T/F508del were selected. The 3849+10kbC-T variant is a class V mutation. Carriers of this mutation are characterized by the presence of residual activity of the CFTR protein, with relatively sufficient pancreatic function, and a milder lesion of the bronchopulmonary system in childhood. The progression of the disease is milder and slower than in patients carrying two severe variants [5,18].

In group 3, 100 patients homozygous for the F508del variant were included. The F508del variant, the most common pathogenic allele of the *CFTR* gene in European and Russian populations, is a class II mutation. The clinical presentation in CF patients homozygous for the F508del mutation is characterized by typical severe manifestations of cystic fibrosis with pancreatic insufficiency and severe lung disease [5,18]. 

The average age of CF diagnosis is significantly lower in the group of patients carrying the W1282R variant compared to the group of patients with the genotype 3849+10kbC-T/F508del (*p* < 0.001; Table 2), but it does not differ from the group of patients homozygous for the F508del variant (*p* < 0.05; Table 2).

Patients included in RCFPR were diagnosed either based on clinical manifestation or results of neonatal screening for CF (CF NS), i.e., often before the onset of symptoms. CF NS has been practiced in Russia since 2007. There were no differences between groups 1—with W1282R and severe variant in trans; and group 3—with F508del/F508del genotypes at the age of CF diagnosis based on clinical symptoms before the CF NS program was introduced into healthcare practice (patients born before 2007) (3.98 ± 1.46 and 2.24 ± 0.36, respectively; Table 2), whereas in the group with 3849+10kbC-T/F508del genotype CF diagnosis was established much later (9.01 ± 1.06; *p* < 0.001; Table 2). This may be partly due to the fact that in most patients with variant 3849+10kbC-T, the exocrine function of the pancreas is preserved, the symptoms of bronchopulmonary system damage develop later and are less pronounced, and the sweat chloride levels are lower than in the other two groups (see below), often borderline or even normal, which delays the diagnosis and initiation of therapy in patients of this group [5,18].

According to the results of CF NS, the diagnosis was established significantly earlier in patients with the genotype carrying W1282R and severe variant in trans than in patients with the 3849+10kbC-T/F508del genotype (*p* < 0.05), the differences with patients homozygous for F508del were not significant (*p* > 0.05; Table 2). This is probably due to the fact that among newborns with a false negative first test for immune-reactive trypsinogen (IRT), patients carrying the 3849+10kbC-T mutation were more common (5 out of 13, i.e., 38.5%) compared to patients from the other two groups: in patients with W1282R and severe variant in trans, this proportion was 8.3% (one out of 12; *p* = 0.16), and in the homozygous for F508del variant it was 2.3% (1 of 44; *p* = 0.0016).

Average values of sweat chloride levels in patients with genotype W1282R and severe variant in trans are significantly higher than in patients with the 3849+10kbC-T/F508del genotype (*p* < 0.0001), but they do not differ from the group of patients homozygous for the F508del variant (*p* > 0.05; Table 3), which might indicate a greater effect on chloride channel function in the case of the W1282R variant compared to the 3849+10kbC-T variant, at least in the sweat glands.

The status of pancreatic function was assessed by the level of fecal elastase-1 (FE). A level of FE less than 200 μg/g indicates a lack of exocrine pancreas function. The level of fecal elastase-1 was reduced in 85.7% of the patients with genotype W1282R and a severe variant in trans. The proportion of pancreatic insufficiency (PI) in patients with the genotype W1282R and severe variant in trans is significantly higher than in patients with the genotype 3849+10kbC-T/F508del (11.1%; *p* < 0.001), and it does not differ when compared with the group of patients homozygous for the F508del variant (91.1%; Table 4).

Nutritional status was assessed using the body mass index (BMI). There were no significant differences in average BMI values between groups of patients with different genotypes when comparing both without considering age and when comparing children separately (under 18 years) and adults (Table 3).

Lung function was assessed by the forced expiratory volume per 1 s (FEV_1_) and forced vital capacity (FVC) of the lungs. In patients with genotype W1282R and severe variant in trans, the average value of FEV_1_ and FVC is slightly lower than in patients with the F508del/F508del genotype (these differences are significant for FVC, *p* < 0.05; Table 3) and do not significantly differ from lung function indexes in patients with the 3849+10kbC-T/F508del genotype (Table 3). The average age of patients in whom FEV_1_ and FVC were measured did not differ in groups with different genotypes (Table 2). The proportion of patients with normal lung function characteristics (≥80% of predicted) in the group with the genotype carrying W1282R and a severe variant in trans is significantly less than in patients with theF508del/F508del genotype (21.4% vs. 57.8%; *p* = 0.0303; Table 4).

Analysis of chronic colonization of the bronchopulmonary system by microorganisms (*Staphylococcus aureus*, *Pseudomonas aeruginosa*, *Burkholderia cepacia*, *Achromobacter xylosoxidans*, *Stenotrophomonas maltophilia*, non-tuberculosis mycobacteria, and Gram-negative microflora) did not reveal significant differences between the compared groups of patients.

Comparison of disease complications did not reveal statistically significant differences in the incidence of the pseudo-Bartter syndrome, osteoporosis, meconium ileus in the anamnesis, sinus polyposis, CF-related diabetes (CFRD), or liver lesions in the four groups (Table 4).

Additionally, the clinical course of the disease was compared in 31 patients who were compound heterozygotes for the W1282R variant and class I or II variant (“severe”) (group 1) and in nine patients carrying the W1282Rvariant in trans with pathogenic *CFTR* gene variants associated with residual pancreatic function (“mild”) (group 4): E92K; L138ins; R334W; D1152H; 3849+10kbC-T; and 4382delA). According to the CFTR2 database, the proportion of patients with residual pancreatic function was 47% for the E92K carriers, 57% for L138ins, 60% for R334W, 76% for D1152H, 67% for 3849+10kbC-T, and 62% for 4382delA [5]. The average age of diagnosis before the introduction of CF neonatal screening is higher in group 4, W1282R and “mild” variant in trans, (19.50 ± 6.11 vs. 3.98 ± 1.46; *p* = 0.0033), which may indicate a milder course of the disease in childhood and, possibly, later age of manifestation of clinical symptoms. This is consistent with the observation of relative sufficiency of the function of the CFTR chloride channel in carriers of “mild” alleles, even in a compound with a “severe” variant [1,2,3,4,16]. No significant differences were found for other indicators, which may be due to the small size of group 4 (only nine persons).

Thus, compared with patients with the 3849+10kbC-T/F508del genotype (group 2), patients with W1282R carrying F508del or class I variants in trans (group 1) were diagnosed earlier based both on clinical manifestation and on the neonatal screening program, which may be due to higher levels in the sweat chloride tests in group 1. The proportion of patients with pancreatic insufficiency was significantly higher in group 1 compared to group 2.

Age of CF diagnosis, sweat chloride level, and pancreatic and nutritional status in patients carrying the W1282R variant and F508del or a class I variant in trans (group 1) did not differ from patients homozygous for the F508del variant (group 3). Respiratory function assessed by FEV_1_ and FVC was slightly better in group 3 compared to group 1 patients.

## 4. Conclusions

The phenotypic consequences of the c.3844T>C (p.Trp1282Arg, W1282R) variant, which is a thymidine–cytosine substitution in exon 22 of the *CFTR* gene, leading to the change of tryptophan-1282 by arginine in the second nucleotide-binding domain (NBD2) of the protein molecule, were described for the first time based on the total sample of 42 CF patients from the Russian Federation.

According to modeling using bioinformatic resources for predicting the functional significance of missense variants, the pathogenicity of the W1282R variant is due to the substitution of an uncharged hydrophobic amino-acid residue by a charged hydrophilic one, and thus the expansion of cavity volume. Analysis of genotype–phenotype features in patients carrying the W1282R variant showed that it can be considered as clinically significant, causing severe CF with pancreatic insufficiency and a significant decrease in lung function. The further functional analysis would allow characterizing the function effect of the W1282R variant more accurately and clarifying its attribution to a certain *CFTR* mutation class. It would also be useful in developing targeted therapy approaches for patients who carry this variant, which is relatively common in Russia.

## Figures and Tables

**Figure 1 genes-11-01137-f001:**
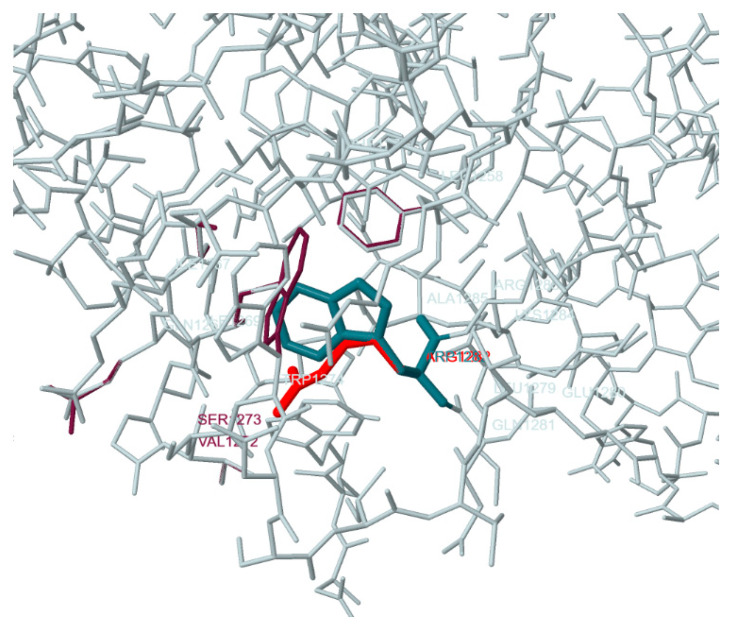
3D simulation of a p.Trp1282Arg (W1282R)-CFTR protein molecule compared to a wild-type molecule. The dark blue color indicates the Trp-1282 residue, and the red color indicates the Arg-1282 residue.

**Table 1 genes-11-01137-t001:** The second alleles of CF patients carrying the c.3844T>C (p.Trp1282Arg, W1282R) variant *in trans*.

Functional Class	Legacy Name	cDNA Name	Protein Name	No. of Patients
II	F508del	c.1521_1523delCTT	p.Phe508del	19
I	CFTRdele2,3	c.54-5940_273+10250del21kb	p.Ser18Arg*fsX16	4
V	3849+10kbC-T	c.3718-2477C>T	p.spl	
I	580-1G>T	c.712-1G->T	p.spl	2
IV	R334W	c.1000C>T	p.Arg334Trp	2
I	W1282X	c.3846G>A	p.Trp1282X	2
II-III	E92K	c.274G>A	p.Glu92Lys	2
I	2184insA	c.2052_2053insA	p.Gln685ThrfsX4	1
I	3821delT	c.3691delT	p.Ser1231ProfsX4	1
n.d. ^1^	4382delA	c.4251delA	p.Glu1418ArgfsX14	1
I	1898+1G>A	c.1766+1G>A	p.spl	1
I	541delC	c.409delC	p.Leu137SerfsX16	1
III	D1152H	c.3454G>C	p.Asp1152His	1
n.d. ^1^	L138ins	c.413_415dupTAC	p.Leu138dup	1
II	R1066C	c.3196C>T	p.Arg1066Cys	1
III	L1335P	c.4004T>C	p.Leu1335Pro	1

^1^ n.d.—not determined.

**Table 2 genes-11-01137-t002:** Mean age of clinical functional manifestation of CF in patients with various genotypes.

Mean Age (Years)	Groups (Genotypes)			
	*1*	*2*	*3*	*4*
at last visit to clinic	13.48 ± 1.67 (0.73–30.71); 31	16.18 ± 1.23 (0.60–32.60); 50	12.94 ± 0.90 (0.53–31.07); 100	23.44 ± 6.67 (0.74–60.06); 9
		*p* = 0.224	*p* = 0.314	*p* = 0.23
at diagnosis	2.74 ± 0.88 (0.075–23.00); 31	9.01 ± 1.06 (0.10–27.78); 50	2.24 ± 0.36 (0.00–15.35); 99	13.07 ± 5.09 (0.12–39.60); 9
		*p* = **0.000019**	*p* = 0.437	*p* = *0.052*
at diagnosis before neonatal screening	3.98 ± 1.46 (0.19–23.00); 17	11.40 ± 1.18 (0.63–27.78); 37	3.65 ± 0.57 (0.00–15.35); 55	19.50 ± 6.11 (4.49–39.60); 6
		*p* = **0.00033**	*p* = 0.31	*p* = **0.0033**
at diagnosis by neonatal screening	0.27 ± 0.13 (0.075–1.51); 11	2.38 ± 0.70 (0.10–7.52); 12	0.46 ± 0.13 (0.008–3.88); 44	0.20 ± 0.05 (0.11–0.29); 3
		*p* = **0.015**	*p* = 0.37	*p* = 0.35
with chronic *Ps. aeruginosa*	17.76 ± 2.38 (1.10–30.71); 15	19.55 ± 1.43 (2.88–32.60); 26	16.14 ± 1.39 (1.38–30.97); 38	32.16 ± 8.10 (13.0–60.06); 5
		*p* = 0.64	*p* = 0.47	*p* = 0.088
without chronic *Ps. aeruginosa*	9.47 ± 1.92 (0.73–23.00); 16	11.82 ± 1.72 (0.60–29.06); 23	10.78 ± 1.12 (0.53–31.07); 61	12.54 ± 9.29 (0.74–40.24); 4
		*p* = 0.12	*p* = 0.70	*p* = 0.71
respiratory function (from sample)	17.45 ± 1.64 (6.19–29.78); 18	18.31 ± 1.41 (6.73–32.60); 24	16.85 ± 0.88 (5.12–30.97); 54	29.59 ± 6.91 (6.14–60.06); 7
		*p* = 0.56	*p* = 0.86	*p* = 0.13
respiratory function (total from RCFPR)	17.45 ± 1.64 (6.19–29.78); 18	25.22 ± 1.61 (6.73–49.89); 42	14.39 ± 0.34 (5.03–42.35); 430	
		*p* = **0.0067**	*p* = *0.051*	

Notes: group 1–W1282Rand severe variant *in trans*; group 2–3849+10kbC-T/F508del; group 3–F508del/F508del; group 4–W1282R and mild variant *in trans*; *p* < 0.05 (marked with bold); *p* < 0.1 (marked with italic).

**Table 3 genes-11-01137-t003:** Mean values of clinical and functional characteristics in CF patients with various genotypes.

Mean Values	Groups (Genotypes)			
	1	2	3	4
Sweat chlorides (mmol/L)	106.82 ± 4.38 (62.0–160.0); 29	80.84 ± 3.22 (52.0–117.0); 40	105.02 ± 2.12 (50.0–157.0); 91	104.00 ± 20.46 (52.0–160.0); 5
		*p* = **0.000031**	*p* = 0.78	*p* = 1.00
BMI (all) (kg/m^2^)	16.41 ± 0.53 (12.24–26.20); 30	16.79 ± 0.33 (13.00–22.00); 50	16.61 ± 0.25 (11.80–24.80); 96	17.60 ± 1.36 (13.15–23.88); 8
		*p* = 0.39	*p* = 0.53	*p* = 0.56
BMI (children) (kg/m^2^)	15.42 ± 0.52 (12.24–20.20); 20	16.32 ± 0.43 (13.00–22.00); 25	15.98 ± 0.25 (11.80–21.60); 73	15.61 ± 0.26 (15.31–16.44); 3
		*p* = 0.14	*p* = 0.29	*p* = 0.53
BMI (adults) (kg/m^2^)	18.37 ± 0.97 (15.80–26.20); 10	17.26 ± 0.49 (13.30–21.70); 25	18.57 ± 0.52 (14.90–24.80); 19	18.74 ± 2.06 (13.15–23.88); 5
		*p* = 0.43	*p* = 0.31	*p* = 0.90
FEV_1_ (% predicted)	64.54 ± 7.01 (16.76–102.50); 18	54.35 ± 4.32 (21.60–86.30); 24	79.64 ± 3.72 (27.20–142.48); 54	62.13 ± 13.41 (17.60–105.30); 7
		*p* = 0.35	*p* = *0.095*	*p* = 0.71
FEV_1_ (% predicted) (total from RCFPR)	64.54 ± 7.01 (16.76–102.50); 18	54.05 ± 3.29 (18.30–87.90); 42	76.93 ± 1.15 (15.00–142.48); 430	
		*p* = 0.23	*p* = *0.064*	
FVC (% predicted)	76.79 ± 4.66 (40.38–117.00); 18	69.27 ± 4.96 (22.70–105.80); 24	89.69 ± 3.14 (43.30–148.35); 53	69.01 ± 10.47 (26.10–102.20); 7
		*p* = 0.32	*p* = **0.026**	*p* = 0.69
FVC (% predicted) (total from RCFPR)	64.54 ± 7.01 (16.76–102.50); 18	70.38 ± 3.71 (21.80–106.00); 42	86.20 ± 1.02 (24.70–176.20); 427	
		*p* = 0.21	*p* = **0.047**	

Notes: group 1–W1282R and severe variant in trans; group 2–3849+10kbC-T/F508del; group 3–F508del/F508del; group 4–W1282R and mild variant in trans; *p* < 0.05 (marked with bold); *p* < 0.1 (marked with italic).

**Table 4 genes-11-01137-t004:** Clinical characteristics of CF patients with various genotypes.

Clinical Features		*Groups*							
		*1*		*2*		*3*		*4*	
		n/N	%	n/N	%	n/N	%	n/N	%
normal RF (FVC&FEV_1_ > 80% pr.)		4/18	22.2	2/24	8.3	29/54	53.7	3/7	42.8
				*p* = 0.37		***p*** = **0.0286**		*p* = 0.355	
First or second-degree decrease of RF (FVC&FEV_1_ 40–79.9% pr.)		9/18	50.0	16/24	66.7	21/54	38.9	2/7	28.6
				*p* = 0.34		***p*** = 0.42		*p* = 0.231	
Third-degree decrease of RF (FVC&FEV_1_ < 40% pr.)		5/18	27.8	6/24	25.0	4/54	7.4	2/7	28.6
				*p* = 1.00		***p*** = **0.0378**		*p* = 1.00	
Fecal elastase 1	>200 µg/g	2	14.3	16	88.9	4	8.9	3	60.0
	<200 µg/g	12	85.7	2	11.1	41	91.1	2	40.0
				***p*** = **0.0001**		*p* = 0.62		*p* = 0.084	
Microflora in anamnesis	*S. aureus*	16/31	51.6	20/49	40.8	53/100	53.0	2/9	22.2
				*p* = 0.36		*p* = 1.00		*p* = 0.149	
	Non-tuberculous *Mycobacteria (NTM)*	0/22	0.0	3/35	8.6	1/74	1.4	0/7	0.0
				*p* = 0.28		*p* = 1.00			
	*Achromobacter*	0/30	0.0	2/48	4.2	8/96	8.3	1/9	11.1
				*p* = 0.52		*p* = 0.19		*p* = 0.23	
	*St. maltophilia*	1/31	3.2	1/49	2.0	3/99	3.0	0/9	0.0
				*p* = 1.00		*p* = 1.00		*p* = 1.00	
	*B. cepacia*	5/31	16.1	3/49	6.1	7/100	7.0	1/9	11.1
				*p* = 0.25		*p* = 0.14		*p* = 1.00	
	Gram-negative flora	4/30	13.3	3/50	6.0	10/100	10.0	1/9	11.1
				*p* = 0.42		*p* = 0.73		*p* = 1.00	
*p. aeruginosa*	chronic	15/31	48.4	26/49	53.1	38/99	38.4	5/9	55.6
				*p* = 0.82		*p* = 0.40		*p* = 1.00	
	chronic: children	7/20	35.0	8/25	32.0	23/70	31.5	1/4	25.0
				*p* = 1.00		*p* = 1.00		*p* = 1.00	
	chronic: adults	8/11	72.7	18/24	75.0	15/26	57.7	4/5	75.0
				*p* = 1.00		*p* = 0.476		*p* = 1.00	
Allergic bronchopulmonary aspergillosis (ABPA)		1/31	3.2	1/50	2.0	0/100	0.0	1/9	11.1
				*p* = 1.00		*p* = 0.236		*p* = 0.403	
Nasal polyposis		5/25	20.0	7/40	17.5	24/89	26.9	3/7	42.9
				*p* = 1.00		*p* = 0.61		*p* = 0.326	
Meconium ileus		3/31	9.7	0/50	0.0	6/90	6.7	1/8	12.5
				*p* = 0.053		*p* = 0.355		*p* = 1.00	
Electrolyte disorder (pseudo-Bartter syndrome, Salt loss syndrome)		1/28	3.6	0/48	0.0	2/98	2.0	0/8	0.0
				*p* = 0.36		*p* = 0.53		*p* = 0.389	
Osteoporosis		3/25	12.0	7/36	19.4	7/69	10.1	3/6	50.0
				*p* = 0.50		*p* = 0.72		*p* = 0.068	
Diabetes		1/31	3.2	0/50	0.0	4/100	4.0	1/9	11.1
				*p* = 0.382		*p* = 1.00		*p* = 0.403	
Liver disease	All	4/31	12.9	3/49	6.1	29/100	29.0	2/9	22.2
				*p* = 0.421		*p* = 0.096		*p* = 0.602	
	cirrhosis with hypertension	1	25.0	0	0.0	9	31.0	0	0.0
	without cirrhosis	3	75.0	3	100.0	20	69.0	2	100.0
				*p* = 1.00		*p* = 1.00		*p* = 1.00	

Notes: group 1—W1282R and severe variant *in trans*; group 2—3849+10kbC-T/F508del; group 3—F508del/F508del; group 4—W1282R and mild variant *in trans*; *p* < 0.05 (marked with bold); pr.—predicted; n—number of patients having the clinical feature; N—number of patients in which the clinical feature was evaluated; %—the proportion of patients having the studied clinical feature.

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
