# Peer review of "Clinical Presentation of the c.3844T>C (p.Trp1282Arg, W1282R) Variant in Russian Cystic Fibrosis Patients"

_genes, 2020, doi:10.3390/genes11101137_

Round 1

Reviewer 1 Report

Petrova et al. Study a series of clinical parameters of CF patients with the W1282R mutation, based on their own results and on data deposited in the RCFPR. The manuscript is interesting and could be important in the management of patients with the W1282R mutation, which is one of the most frequent in some regions of the Russian Federation.

However, the article is extremely difficult to read. The use of three nomenclatures to mention each mutation causes the thread of reading to be lost, forcing us to reread each passage. I suggest that after the first mention of each mutation, it was said by a single nomenclature.

The definition of groups of mutations is not clear. It requires a further explanation of the rationale to decide why a certain mutation in the second allele falls into each of the groups. It would be very useful to redo table 1 with the second mutation in a separate column for each group to make it more readable, and to have a good reference when reading the text.

Minor issues

The acronym RCFPR should not be used in the abstract.

The mention of the statistical program should contain the commercial house that produces it, or a reference.

Figure 1 is useless, unless you do not want to give a more detailed explanation of the result.

Tables 2 and 3 could be replaced by box-plots, and put the tables as supplementary data.

Table 4 lacks explanation. It is not known what units are used, or what the percentage means. I guess they are the percentage of patients with a certain characteristic. But, for Ostroporosis or Batter's syndrome there is no explanation of what are considered normal values, and how they are established.

Author Response

Petrova et al. Study a series of clinical parameters of CF patients with the W1282R mutation, based on their own results and on data deposited in the RCFPR. The manuscript is interesting and could be important in the management of patients with the W1282R mutation, which is one of the most frequent in some regions of the Russian Federation.

However, the article is extremely difficult to read. The use of three nomenclatures to mention each mutation causes the thread of reading to be lost, forcing us to reread each passage. I suggest that after the first mention of each mutation, it was said by a single nomenclature.

Response: Dear Reviewer, thank you for your comments. We agree and used the full name of pathogenic variants (legacy, cDNA, and protein name) only once in the Introduction (lines 56-62, page  2) and in table 1 (lines 186-188, pages 5-6). In the following text, we use the legacy names of the CFTR variants now.

The definition of groups of mutations is not clear. It requires a further explanation of the rationale to decide why a certain mutation in the second allele falls into each of the groups.

Response: We changed paragraph to:

Phenotypic correlations were studied in a group of patients carrying the W1282R variant (group 1) compared to two groups of patients with different genotypes whose clinical presentation is well known and significantly different. In group 1 there were 31 patients carrying the W1282R and class I or II variant in trans (F508del, CFTRdele2.3, 1898+1G->A, 2184insA, 3821delT, 541delC, 712-1G->T, or W1282X) (table 1).

In general, patients homozygous for class I–III mutations exhibit a phenotype associated with pancreatic insufficiency, higher frequency of meconium ileus, premature mortality, earlier and more severe deterioration of lung function, higher incidence of malnutrition and severe liver disease. Class IV–V mutations are usually associated with milder lung disease, older age at death, pancreatic sufficiency. Class IV–V mutations are phenotypically dominant when occurring in combination with class I–III mutations [26].

For group 2 50 patients with compound heterozygous genotype 3849+10kbC-T/F508del were selected. The 3849+10kbC-T variant is a class V mutation. Carriers of this mutation are characterized by the presence of residual activity of the CFTR protein, with relatively sufficient pancreatic function, and a milder lesion of the bronchopulmonary system in childhood. The progression of the severity of the disease is milder and slower than in patients carrying two severe variants [5, 18].

In group 3 100 patients homozygous for the F508del variant were included. The F508del variant, the most common pathogenic allele of the CFTR gene in European and Russian populations, is a class II mutation. The clinical presentation in CF patients homozygous for the F508del mutation is characterized by typical severe manifestations of cystic fibrosis with pancreatic insufficiency and severe lung disease [5, 18].

It would be very useful to redo table 1 with the second mutation in a separate column for each group to make it more readable, and to have a good reference when reading the text.

Response: We agree and changed table 1 (lines 186-188, pages 5-6)

Minor issues

The acronym RCFPR should not be used in the abstract.

Response: We changed RCFPR to CF Patients Registry (Abstract page 1)

The mention of the statistical program should contain the commercial house that produces it, or a reference.

Response: Added to lines 136-137, page 3: (StatSoft. Inc., USA)

Figure 1 is useless, unless you do not want to give a more detailed explanation of the result.

Response: We removed fig.1 from the text.

Tables 2 and 3 could be replaced by box-plots, and put the tables as supplementary data.

Response: We would prefer to leave the presentation of the material in the form of tables 2 and 3, because the presentation in the form of diagrams will complicate the understanding of the material.

Table 4 lacks explanation. It is not known what units are used, or what the percentage means. I guess they are the percentage of patients with a certain characteristic.

Response:  Added to Notes of Table 4 “n – number of patients having the clinical feature; N – number of patients in which the clinical feature was evaluated; % - proportion of patients having the studied clinical feature”.

But, for Ostroporosis or Batter's syndrome there is no explanation of what are considered normal values, and how they are established.

Response:  Added to Methods: Diagnostic criteria of manifestations and complications of CF were used according to the Variables and Definitions of the ECFS Patient Registry (https://www.ecfs.eu/projects/ecfs-patient-registry/Variables-Definitions). Lines-112-114

Reviewer 2 Report

Petrova, et al conducted a retrospective chart review in combination with extensive genotyping in an effort to assess the clinical presentation of a CFTR-causing mutation seen in Russia, i.e. W1282R. The authors compared the various clinical parameters between patients carrying W1282R/a severe CF-causing mutation (group 1) and, either F508del/3849+10kbC-T (a mild CF-causing mutation) (group 2), or F508del/F508del (group 3). Based on the analyses, W1282R appears to cause at least as severe CF symptoms as F508del homozygotes as reflected in the age of diagnosis, sweat chloride concentration, pancreatic function, and lung function. In contrast, the 3849+10kbC-T mutation causes milder clinical phenotypes. This is the first systematic analysis of the clinical phenotypes of the W1282R mutation, and the PCR based restriction digestion assay might be a useful clinical diagnosis or even newborn screening for CF in Russia.

The abstract can be significantly improved by highlighting the key findings of the study. The genotype expression can be simplified to facilitate reading. RCFPR needs to be spelled out before the acronym is used. Some of the discussion in the Conclusion section should be moved to the previous section.

Finally, the authors have not mentioned if they have done full-length CFTR gene sequencing for patient alleles carrying the W1282R mutation. This is important to rule out possible presence of additional CF-causing mutation in those alleles. Given the extensive genotyping done in this study, this reviewer would assume that the authors have such information.

Author Response

The abstract can be significantly improved by highlighting the key findings of the study.  

Response: Dear Reviewer, Thank you for your comments,

We have changed the Abstract (page 1)

Background: The goal was to study the phenotypic manifestations of c.3844T>C (p.Trp1282Arg, W1282R) variant, a CF-causing mutation in Russia. Materials and Methods: Clinical manifestations and complications (the age at CF diagnosis, sweat test, pancreatic status, lung function, microbial infection, body mass index (BMI), the presence of meconium ileus (MI), diabetes, and severe liver disease) were compared in four groups: group 1 –patients carrying c.3844T>C  and severe class I or II variant in trans; group 2 – 3849+10kbC>T/F508del patients; group 3 – F508del/F508del patients; and group 4 –patients with W1282R and ”mild” variant in trans. Results: Based on the analyses, W1282R with class I or II variant in trans appears to cause at least as severe CF symptoms as F508del homozygotes as reflected in the early age of diagnosis, high sweat chloride concentration, insufficient pancreatic function, and low lung function, in contrast to 3849+10kbC-T/F508del compound heterozygotes having milder clinical phenotypes. Conclusion: The W1282R pathogenic variant, is seemed to lead to severe disease phenotype with pancreatic insufficiency, similarly to the F508del homozygous genotype.

The genotype expression can be simplified to facilitate reading

Response: The full name of pathogenic variants (legacy, cDNA, and protein name) was specified once in Introduction (lines 56-2, page  2) and in table 1 (lines 186-188, pages 5-6). In the following text, we use the legacy names of the CFTR variants.

RCFPR needs to be spelled out before the acronym is used.

Response: We agree and removed  RCFPR from the Abstract (Abstract, page 1).

Some of the discussion in the Conclusion section should be moved to the previous section.

Response: We moved 2 paragraphs to Results and discussion (to lines 329-337, page 10).

 Finally, the authors have not mentioned if they have done full-length CFTR gene sequencing for patient alleles carrying the W1282R mutation. This is important to rule out possible presence of additional CF-causing mutation in those alleles. Given the extensive genotyping done in this study, this reviewer would assume that the authors have such information.

Response: The absence of other pathogenic variants in cis with W1282R was confirmed by sequencing the coding sequence of the CFTR gene and searching for large rearrangements using the MLPA method. (Added to Methods lines 100-102).

Round 2

Reviewer 1 Report

Authors have accepted most of my comments and corrected the text consequently. The article merits to be published in its actual form.